# Genomics-Enabled Management of Genetic Resources in Radiata Pine

**Jaroslav Klápště [1]**, **Ahmed Ismael [1,2]**, **Mark Paget [3]**, **Natalie J. Graham [1]**, **Grahame T. Stovold [1]**, **Heidi S. Dungey [1]** **and Gancho T. Slavov [1,\*]**

[1] Scion (New Zealand Forest Research Institute Ltd.), Private Bag 3020, Rotorua 3010, New Zealand; Jaroslav.Klapste@scionresearch.com (J.K.); ahmedismaelsayed@gmail.com (A.I.); Natalie.Graham@scionresearch.com (N.J.G.); Toby.Stovold@scionresearch.com (G.T.S.); Heidi.Dungey@scionresearch.com (H.S.D.)

[2] Research and Development, Livestock Improvement Corporation, Private Bag 3016, Hamilton 3240, New Zealand

[3] Radiata Pine Breeding Company, 99 Sala Street, Rotorua 3010, New Zealand; mark.paget@rpbc.co.nz

[\*] Correspondence: gancho.slavov@scionresearch.com

**Abstract:** Traditional tree improvement is cumbersome and costly. Our main objective was to assess the extent to which genomic data can currently accelerate and improve decision making in this field. We used diameter at breast height (DBH) and wood density (WD) data for 4430 tree genotypes and single-nucleotide polymorphism (SNP) data for 2446 tree genotypes. Pedigree reconstruction was performed using a combination of maximum likelihood parentage assignment and matching based on identity-by-state (IBS) similarity. In addition, we used best linear unbiased prediction (BLUP) methods to predict phenotypes using SNP markers (GBLUP), recorded pedigree information (ABLUP), and single-step "blended" BLUP (HBLUP) combining SNP and pedigree information. We substantially improved the accuracy of pedigree records, resolving the inconsistent parental information of 506 tree genotypes. This led to substantially increased predictive ability (i.e., by up to 87%) in HBLUP analyses compared to a baseline from ABLUP. Genomic prediction was possible across populations and within previously untested families with moderately large training populations ($N$ = 800–1200 tree genotypes) and using as few as 2000–5000 SNP markers. HBLUP was generally more effective than traditional ABLUP approaches, particularly after dealing appropriately with pedigree uncertainties. Our study provides evidence that genome-wide marker data can significantly enhance tree improvement. The operational implementation of genomic selection has started in radiata pine breeding in New Zealand, but further reductions in DNA extraction and genotyping costs may be required to realise the full potential of this approach.

**Keywords:** tree breeding; pedigree reconstruction; genomic selection; genomic prediction; single-step BLUP; *Pinus radiata*

## 1. Introduction

Despite the increasing appreciation of deforestation as a major environmental threat, global rates of forest loss have not decreased and continue to be largely driven by anthropogenic changes in land use [1–3]. This trend will likely be exacerbated by projected human population growth and climate change, both of which are expected to further aggravate the looming environmental crisis and put significant pressure on the long-term sustainability of natural renewable resources such as wood fibre [4]. Thus, the forest-based "circular" bioeconomy model, which is based on wood and products derived from wood as a dominant part of the system [5], is facing the massive challenge of growing increasing amounts of wood biomass under ever-increasing pressure to convert forests to agricultural land [6].

Traditional tree breeding is slow, expensive, and complex as the choice of species, balance of genetic gain versus diversity, and scale of genetic testing and conservation

should be considered [7,8]. Climate change is further increasing this complexity, and assisted migration is increasingly recognised as a potentially effective climate change adaptation tool at the population level [9–12], while the importance of within-population adaptive genetic variation is also appreciated [13].

The development of genomic resources in forest tree species has enabled (1) efficient population management in both conservation and breeding programmes (e.g., this Special Issue); (2) effective tracing of co-ancestry and inbreeding [14,15]; (3) dissection of the genetic architecture of complex traits through genome-wide association studies (GWAS) to identify candidate genes for economically important or adaptive traits [16–19]; and (4) whole-genome regression modelling to predict unobserved phenotypes and select genetically superior individuals without field testing (i.e., genomic selection, [20–23]). Because forest tree breeding is a long process, historical material is often unavailable for genotyping. Thus, approaches combining data for genotyped and non-genotyped individuals are potentially very useful. Blended "single-step" models combine phenotypic, pedigree, and genomic data, which makes it possible to use historical records and reduces the burden of genotyping costs in operational schemes [24]. However, there are still significant technical challenges with this approach. For example, the marker-based relationship matrix needs to be rescaled appropriately relative to the pedigree-based relationship matrix to avoid bias in genomic breeding value estimates [25,26]. Furthermore, the relative weighting of the pedigree versus genomic information needs to be assessed carefully during the matrix blending process to prevent matrix inversion issues [25,26].

In this study, we used an extensive genomic and phenotypic data set from New Zealand's radiata pine breeding programme to (1) objectively characterise population structure, (2) perform large-scale pedigree reconstruction, and (3) assess the potential for operational implementation of genomic or single-step blended prediction.

## 2. Materials and Methods

### 2.1. Plant Materials and Phenotypic Data

Radiata pine occurs naturally in five locations (provenances): Año Nuevo (CA, USA), Monterey (CA, USA), Cambria (CA, USA), Cedros Island (Mexico), and Guadalupe Island (Mexico, Figure 1). The species was first introduced to New Zealand ca. 170 years ago and has shown excellent growth under the local environmental conditions. Populations of ca. 1000 individuals were collected from forest stands across New Zealand during 1950–1988. This included trees from all five provenances growing in provenance tests established in New Zealand. Several sublines were established targeting different traits such as growth and form, wood density, Dothistroma needle blight resistance, and internode length. However, after the most recent revision of the breeding programme, a nucleus breeding strategy was proposed [27]. This strategy featured a large open-pollinated main population and a smaller elite population, with 50% of testing via progeny tests and 50% via clonal tests. Overall, more than 400,000 trees have been grown and measured in more than 100 replicated field trials.

We used phenotypic and, where available, genomic data for a subset of 4254 tree genotypes from five historical populations ("260", "313", "314", "397", and "399") tested in five trial series ("FR260", "FR305GF", "FR305HD", "FR353", and "Cloned Elites", Table 1 and Supplementary Materials Table S1) for genomic and single-step blended prediction analyses. We focused on two phenotypic traits, both of which are part of the breeding objective for radiata pine in New Zealand: diameter at breast height (DBH), which was measured using a diameter tape at tree height of 1.4 m, and wood density (WD), which was estimated through the maximum moisture content method [28]. In addition to material directly involved in the breeding programme, we also used two biparental linkage mapping populations (Table 1) to test whether genomic prediction is effective within families. Thus, we used phenotypic data for 4430 tree genotypes (4254 from historical populations as described above and 176 from the two mapping families) and genomic data for 2466 tree genotypes (Table 1).

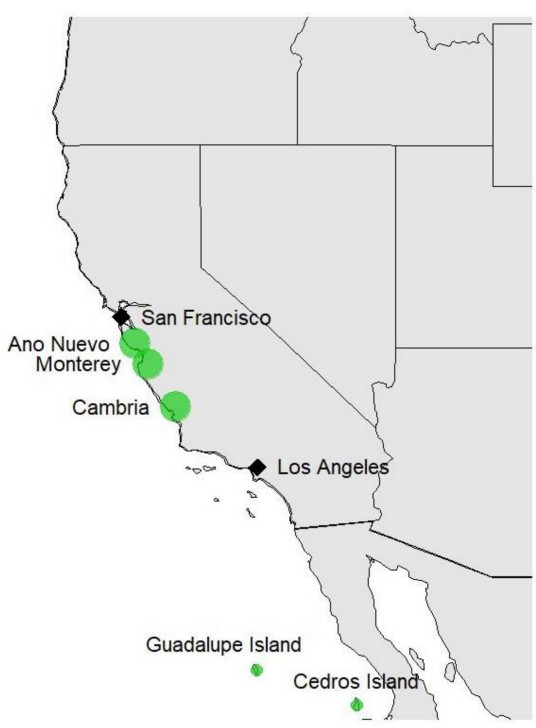

**Figure 1.** Natural distribution of radiata pine in North America.

**Table 1.** Populations, families, and trial series included in genomic (GBLUP), pedigree-based (ABLUP), and single-step blended (HBLUP) prediction analyses (see Table S1 for field trial details).

| Population | Description | Analysis | Number of Parents (Families) | Number of Genotyped Individuals | Number of Genotypes Used in GBLUP |
|---|---|---|---|---|---|
| 260 | Progeny test | ABLUP, HBLUP | 47 (26) | 0 | 0 |
| 313 | Cloned Elites | ABLUP, GBLUP, HBLUP | 55 (74) | 695 | 681 |
| 314 | Cloned Elites | ABLUP, HBLUP | 55 (75) | 609 | 0 |
| 397 | Older Clonal Tests | ABLUP, GBLUP, HBLUP | 64 (50) | 464 | 444 |
| 399 | Older Clonal Tests | ABLUP, GBLUP, HBLUP | 24 (42) | 522 | 469 |
| QTL | Mapping family (268,405 × 268,345) | GBLUP | 2 (1) | 93 | 86 |
| FWK | Mapping family (850,055 × 850,096) | GBLUP | 2 (1) | 83 | 81 |
| Total | | | | 2466 | 1761 |

### 2.2. Genomic Data

Single-nucleotide polymorphism (SNP) data were generated using two genotyping platforms. First, an exome capture genotyping-by-sequencing (GBS) method [29] was implemented using probes obtained by resequencing transcriptomes from multiple tissues [30,31]. This platform allowed us to generate data for 50,917 SNP markers. Second, a subset of the GBS markers were used to develop a custom radiata pine Affymetrix Axiom 36K array, NZPRAD02, which included 36,285 SNPs [32].

Where necessary, we combined data by identifying a subset of SNPs that performed consistently well using both platforms. Briefly, we genotyped 295 individuals using both GBS and the NZPRAD02 Axiom array, and then we identified 9353 SNPs for which genotypic calls were strongly correlated between the two platforms (Pearson's *r* > 0.9).

*2.3. Data Analysis*

2.3.1. Population Structure

We used the *smartpca* function within the EIGENSTRAT package to perform individual-based principal component analysis [33] of all genotyped individuals. This was done using the set of 9353 SNPs that had good transferability between genotyping platforms (see Section 2.2).

2.3.2. Pedigree Reconstruction

Pedigree reconstruction was performed on 2476 individuals (populations "313", "314", "397", and "399"), with 42 individuals genotyped using the Axiom SNP array and 2434 individuals genotyped using GBS [29,31]. Marker data were combined as described above and then filtered for an individual call rate > 0.75 and SNP call rate > 0.95, leaving 2290 tree genotypes and 8523 SNPs for further analyses.

Pedigree reconstruction analyses then followed five steps. First, we calculated pedigree-based and genomic relationships between trees. Pedigree-based relationship coefficients (APED) were calculated based on information from the Radiata Pine Breeding Company (RPBC) using the *pedigree* R-package [34]. Genomic relationships were estimated using the identity-by-state (IBS) allele-sharing similarity statistic from PLINK 1.9 [35,36]. We used IBS coefficients because they make few population genetic assumptions and are robust to the presence of genetically differentiated groups [37,38]. Second, we used the two linkage mapping populations (Table 1) to estimate the IBS threshold separating first-degree relatives (full-sib or parent–offspring) from unrelated individuals. This was possible because parent–offspring relationships in these populations had been validated previously using microsatellite markers [39]. Third, we performed a preliminary quality assurance check to identify unexpected duplicates (i.e., samples from presumably unrelated trees with IBS > 0.95) or unexpected conflicts (i.e., samples from the same genotype with IBS < 0.95). We compared all duplicates and conflicts with their presumed parents and offspring based on the pedigree information to resolve ambiguous identities. Unresolved duplicates and conflicts were assigned proxy identities and were also included in downstream analyses. Fourth, we performed computer simulations to assess the effectiveness of the *apparent* R package [40] for pedigree reconstruction. The *apparent* package searches for triplets of individuals (two parents and an offspring) by minimising genetic distance based on dense marker arrays [40]. We tested four scenarios in which the population of candidate parents contained the true parents and (i) 100 randomly selected parents genotyped using GBS; (ii) 100 randomly selected parents genotyped using the Axiom array; (iii) 50 randomly selected parents genotyped using GBS and 50 randomly selected parents genotyped using the Axiom array; or (iv) 50 randomly selected full-sibs (25 from each mapping population), all genotyped using GBS. For each scenario, we used the number of correct assignments and the number of statistically significant assignments as performance criteria. Finally, we performed the actual pedigree reconstruction for the 2290 individuals included in the filtered data set.

2.3.3. Genomic Prediction (GBLUP)

We tested genomic prediction within and across populations and families, and then we identified the minimum number of SNPs required for accurate prediction. This was accomplished using genomic best linear unbiased prediction (GBLUP, [41,42]), the *rrBLUP* R package [43], and the following model:

$$\boldsymbol{y} = \boldsymbol{X\beta} + \boldsymbol{Zu} + \boldsymbol{e} \tag{1}$$

where $\boldsymbol{y}$ is a vector of observed phenotypes, estimated as clonal best linear unbiased predictors (BLUPs) corrected for site and design terms using a mixed linear model fitted using the *lme4* R package [44,45]; $\boldsymbol{\beta}$ is a vector of fixed effects (overall mean); $\boldsymbol{u}$ is a vector of random genomic estimated breeding values following $var(\boldsymbol{u}) \sim N(0, \boldsymbol{G}\sigma_u^2)$, where $\boldsymbol{G}$ is

the marker-based relationship matrix [46] and $\sigma_u^2$ is the additive genetic variance captured by genetic markers; $e$ is a vector of random residual effects following $var(e) \sim N(0, I\sigma_e^2)$, where $I$ is the identity matrix and $\sigma_e^2$ is the residual variance; and $X$ and $Z$ are index matrices assigning the fixed effects in $\beta$ and random effects in $u$ to the phenotypes in $y$.

To assess the effectiveness of GBLUP prediction, we used random *k*-fold cross-validations in which the data were randomly partitioned into training and test sets, or cross-validations in which a particular population or family was entirely or partly removed from the training set [47]. To quantify GBLUP effectiveness, we calculated predictive ability as the correlation between predicted (i.e., GBLUP) and "observed" phenotypes (i.e., clonal BLUPs, see above). We excluded the non-genotyped "260" population, as well as the "314" population because of the large proportion of missing data for WD. We also excluded tree genotypes with missing phenotypic data. Thus, the data set for GBLUP analyses included 1761 of the 1857 tree genotypes from three breeding populations ("313", "397", and "399") and two biparental mapping families ("QTL" and "FWK"; Table 1). In random cross-validations, predictive abilities were calculated based on the observed and predicted values of all 1761 tree genotypes. Only the predicted values for the test set were used, and data were split randomly into *k* folds (i.e., based on the desired size of the training set) until a predicted value was available for each genotype. In cross-validations across populations or within families, predictive abilities were calculated only based on the values for individuals in the target population or family that were not included in the training set in any given iteration of the analysis. For example, when 20 individuals from the "QTL" family were included in the training set, predictive abilities were calculated based on the observed and predicted values of the remaining 66 individuals in this family that were in the test set (Table 1).

### 2.3.4. Pedigree-Based and Single-Step Blended Prediction (ABLUP and HBLUP)

Pedigree prediction (ABLUP) uses the pedigree alone to predict phenotypes, whereas single-step blended prediction (HBLUP) combines pedigree and genomic data. Because the quality of the pedigree affects the resulting prediction accuracy, we compared scenarios of ABLUP and HBLUP using (i) the originally recorded pedigree; (ii) a pedigree corrected through pedigree reconstruction, but with unresolved cases reverting to the originally recorded pedigree; and (iii) a pedigree corrected through pedigree reconstruction, with unresolved cases set as "parent unknown".

The ABLUP and HBLUP linear mixed models were implemented in the *breedR* R package [48] as follows:

$$y = X\beta + Zu + e_w \tag{2}$$

where $y$ is a vector of observed phenotypes (i.e., measurements), $\beta$ is a vector of fixed effects such as site and replication within site, $u$ is a vector of random terms (described below), $X$ and $Z$ are index matrices assigning the fixed effects in $\beta$ and random effects in $u$ to the phenotypes in $y$, and $e_w$ is the vector of weighted residuals following $var(e_w) \sim N(0, I\sigma_{e_w}^2)$, where $I$ is the identity matrix and $\sigma_{e_w}^2$ is the weighted residual variance following $\sigma_{e_w}^2 = average((diag(W)x\sigma_e^2)^{-1})$. $W$ is a diagonal matrix of weights following:

$$W = \begin{pmatrix} w_1 & 0 & \dots & 0 \\ 0 & w_2 & \dots & 0 \\ \vdots & \vdots & \ddots & \vdots \\ 0 & 0 & \dots & w_n \end{pmatrix} \tag{3}$$

where $w_n$ is the weight estimated for the *n*th environment $\left(w_n = \frac{1}{\sigma_n^2}\right)$, with $\sigma_n^2$ being the site-specific residual variance from a site-specific linear regression model, which only includes fixed terms. The random terms in $u$ includes the (i) site-specific effects of blocks within replications $b$, which follows $var(b) \sim N(0, I\sigma_b^2)$, where $I$ is the identity matrix and $\sigma_b^2$ is the site-specific block-within-replication variance; and (ii) additive genetic effects $a$, which follows $var(a) \sim N(0, K\sigma_a^2)$, where $K$ denotes the pedigree-based relationship

matrix $A$ [49] in pedigree-based analyses (ABLUP) or the "blended" relationship matrix $H$ combining information from the pedigree and SNP markers in HBPLUP analyses (described below), and $\sigma_a^2$ is the additive genetic variance.

Marker-based and pedigree-based relationship matrices generally have different scales because they have different reference (base) populations [50]. This can affect the accuracy of predicted genomic breeding values [51]. Therefore, adjusting the marker-based relationship matrix $G$ to the reference population of the pedigree-based relationship matrix $A$ (i.e., making the two matrices equivalent) is the most crucial step in single-step evaluations. The marker-based relationship matrix $G$ was constructed following the approach of VanRaden [46]:

$$G = \frac{ZZ'}{2\sum_j p_j(1 - p_j)} \tag{4}$$

where $Z = M - P$; $M$ is a matrix of marker genotypes coded 0, 1, and 2, corresponding to the number of non-reference alleles; $P$ is a vector of the expected numbers of non-reference alleles (i.e., the doubled frequencies of the alternative alleles); and $p_j$ is the frequency of the non-reference allele at the $j$th locus.

Re-scaling of the marker-based relationship matrix was performed following the approach of Gao et al. [52]:

$$\begin{cases} Avg.diag(G)\beta + \alpha = Avg.diag(A_{22}) \\ Avg.offdiag(G)\beta + \alpha = Avg.offdiag(A_{22}) \end{cases} \tag{5}$$

where $A_{22}$ is the pedigree-based relationship matrix for genotyped individuals. The $G$ matrix is usually not positive semi-definite, which is one of the assumptions of mixed linear models. Therefore, relative weighting ($w$) of the genomic and pedigree-based relationship matrices was used to calculate a weighted $G$ matrix ($G_w$) as follows:

$$G_w = wA_{22} + (1 - w)G \tag{6}$$

The blended $H$ matrix, combining SNP and pedigree-based information, was constructed as follows:

$$H = \begin{bmatrix} A_{11} + A_{12}A_{22}^{-1}(G_w - A_{22})A_{22}^{-1}A_{21} & A_{12}A_{22}^{-1}G_w \\ G_wA_{22}^{-1}A_{21} & G_w \end{bmatrix} \tag{7}$$

where $A_{11}$ is the relationship matrix for non-genotyped individuals, $A_{12}$ and $A_{21}$ are the relationship matrices between genotyped and non-genotyped individuals, and $A_{22}$ is as defined above.

The narrow-sense heritability ($h^2$, Table 2) for each ABLUP and HBLUP scenario was calculated as:

$$h^2 = \frac{\sigma_a^2}{\sigma_a^2 + \sigma_{e_w}^2} \tag{8}$$

Standard errors of variance components and narrow-sense heritability were estimated through the delta method using a first-order Taylor approximation in *breedR*. Cross-validations were performed by including all but one trial series in the training set, with the remaining trial series used as a test set. The performance of ABLUP and HBLUP scenarios was assessed using predictive ability as described above for GBLUP.

**Table 2.** Narrow-sense heritabilities ($h^2$) and their standard errors (in parentheses) for diameter at breast height (DBH) and wood density (WD) from three different pedigree-based models (ABLUP) and three single-step blended prediction models (HBLUP) using the default weight (0.05) on pedigree information.

| Model | Pedigree | Unresolved Parentage | DBH | WD |
|---|---|---|---|---|
| ABLUP | Originally recorded | Originally recorded | 0.248 (0.009) | 0.409 (0.011) |
| ABLUP | Corrected | Originally recorded | 0.245 (0.009) | 0.488 (0.011) |
| ABLUP | Corrected | Unknown | 0.234 (0.009) | 0.467 (0.011) |
| HBLUP | Originally recorded | Originally recorded | 0.186 (0.001) | 0.436 (0.011) |
| HBLUP | Corrected | Originally recorded | 0.213 (0.009) | 0.435 (0.011) |
| HBLUP | Corrected | Unknown | 0.212 (0.009) | 0.434 (0.011) |

## 3. Results

### 3.1. Population Structure

Principal component analysis of the SNP data (Figure 2) clearly separated the two biparental mapping families ("FWK" and "QTL"), but not the four breeding populations, which are strongly inter-connected. For example, the "397" and "399" populations are closely related and contain many of the parents of the "313" and "314" populations.

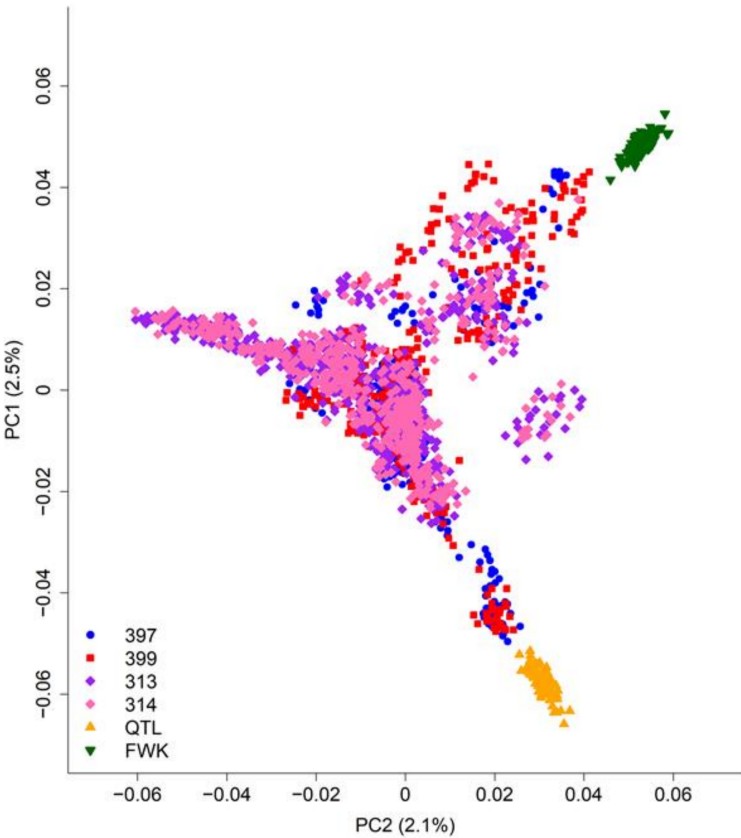

**Figure 2.** Principal component analysis of population structure for all genotyped radiata pine trees (Table 1). The percentages of single-nucleotide polymorphism (SNP) variation captured by the first two principal components (PC1 and PC2) are shown in parentheses.

### 3.2. Pedigree Reconstruction

Identity-by-state (IBS) analysis of the linkage mapping populations showed a clear separation of first-degree relatives from unrelated individuals with an empirical threshold value of 0.85 (Figure 3). This threshold appeared to be consistent across both genotyping

platforms (Figure S1) and allowed us to identify two types of inconsistencies between pedigree-based and SNP-based relationships. The first set of discrepancies consisted of pairs of individuals presumed to be unrelated based on pedigree information but appearing to be closely related based on SNPs (IBS > 0.85). This set consisted of 14,853 cases, or 0.7% of the pairs presumed to be unrelated. The second set consisted of first-degree relatives based on pedigree information (i.e., full-sib or parent–offspring pairs) that appeared unrelated based on SNPs (IBS < 0.85). This set consisted of 2215 cases, or 10% of the presumed first-degree relatives.

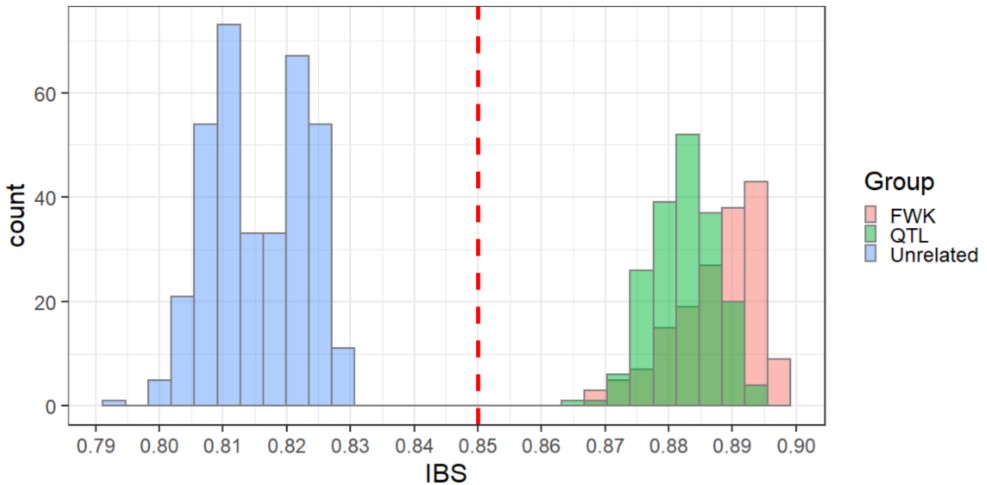

**Figure 3.** Distribution of identity-by-state (IBS) similarity coefficients within and across families. IBS coefficients between parents and their offspring are shown in green ("QTL" family) and red ("FWK" family). IBS coefficients for unrelated trees in the "QTL" versus "FWK" families are shown in blue. An empirical threshold between unrelated and related trees was set at IBS = 0.85 (red line).

For all simulated scenarios, analyses using *apparent* resulted in assigning the true parents for all, or nearly all (≥98%), offspring (Table S2). For scenarios comparing genotyping platforms (Scenarios 1–3), at least 99% of the assignments were statistically significant. In the fourth scenario, which tested the effect of relatedness among candidate parents, the proportion of statistically significant assignments was much lower (51%).

Across the four genotyped breeding populations, 506 individuals had inconsistent parentage based on the recorded pedigree versus SNP data. Of these, 37 had inconsistent maternal information, 404 had inconsistent paternal information, and 65 had inconsistent information for both parents. A combined parentage analysis, which was performed using the *apparent* R package and IBS matching (i.e., IBS > 0.85), identified putative parents for 241 individuals (48%). However, only 51 (21%) of these assignments were statistically significant based on the *apparent* analysis.

### 3.3. Genomic Prediction (GBLUP)

To set a baseline of genomic predictive ability, we first performed random cross-validations in which genotyped individuals (Table 1) were randomly assigned to training and test populations, and then subsets of markers were sampled at random (Figure 4). Predictive abilities were moderate for both DBH and WD but substantially higher for WD. This was expected because of the higher narrow-sense heritabilities for WD (0.41–0.49) than for DBH (0.19–0.25) (Table 2). Furthermore, training set sizes of 800–1200 and as few as 2000 markers appeared to capture most of the predictive ability achievable with this data set.

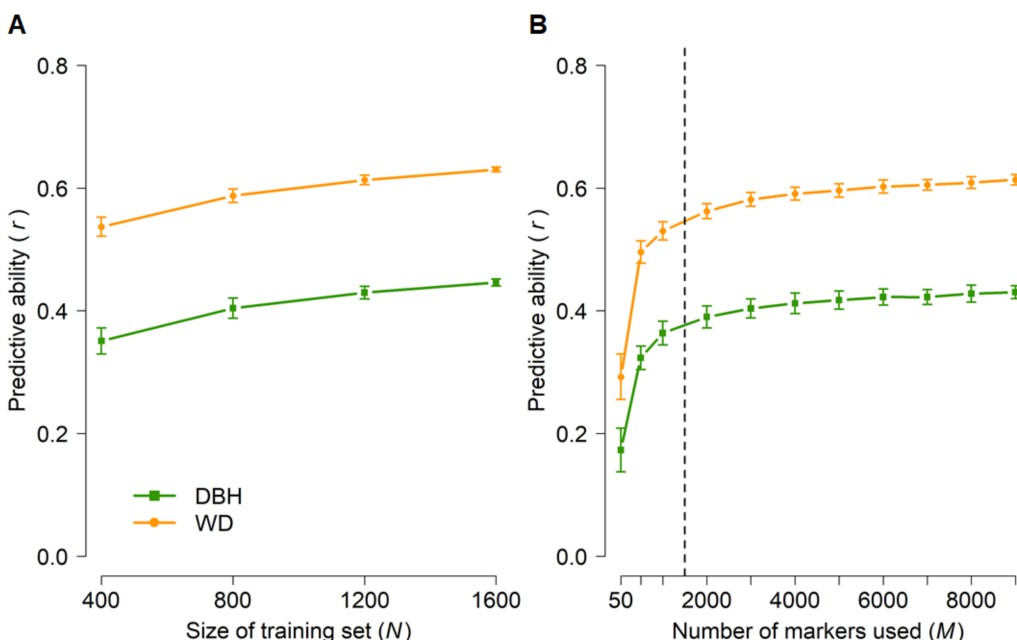

**Figure 4.** Genomic predictive abilities for diameter at breast height (DBH) and wood density (WD) as a function of training set size (**A**) and number of markers (**B**) used in random cross-validations of 1761 radiata pine genotypes from five populations (Table 1). Error bars correspond to standard deviations across 100 random cross-validations for each set of parameters. Analyses in (**A**) were based on all 9353 markers. The training set size in (**B**) was $N = 1200$ and subsets of markers were selected at random. Predictive abilities were calculated based on the predicted and observed values of all 1761 genotypes, after data had been split randomly into $k$ folds until a predicted value was available for each genotype (see Materials and Methods and Table 1). The dashed line in (**B**) indicates the plateau in predictive ability.

Next, we explored scenarios in which the phenotypes of specific populations were predicted from training sets that included a small subset of individuals from the same population. These scenarios were used to simulate reduced field testing. More specifically, we tested whether we could predict the phenotypes of the more advanced "313" population by training a model on the other populations and families included in GBLUP analyses. The average predictive abilities for DBH and WD were 0.17 and 0.41, respectively, when no "313" genotypes were included in the training set (Figure 5A). However, including even as few as 100 (15%) of the "313" genotypes in the training set resulted in roughly 50% higher predictive abilities, with no further improvements when 200 or 300 "313" genotypes were included in the training set. Finally, GBLUP predictions based on as few as 2000 randomly selected markers were nearly as good as those based on the whole set of 9353 markers (Figure 5B).

The relative importance of within-family selection is likely to increase as genomic selection is implemented operationally. Therefore, we tested whether within-family genomic prediction can be effective by applying the across-population cross-validation procedure used for the "313" population (see above) for the "QTL" and "FWK" biparental mapping families (Figure 6). As expected from previous studies, prediction of DBH was poor (predictive abilities $\leq 0.1$). The predictive ability for WD was moderate in both families (0.34–0.52), though including subsets of "QTL" or "FWK" genotypes in the training population did not substantially improve predictive ability (Figure 6A,C). Compared to other scenarios, predictive abilities increased more slowly with the number of markers used: a plateau was reached at $M = 3000$–5000 SNPs (Figure 6B,D).

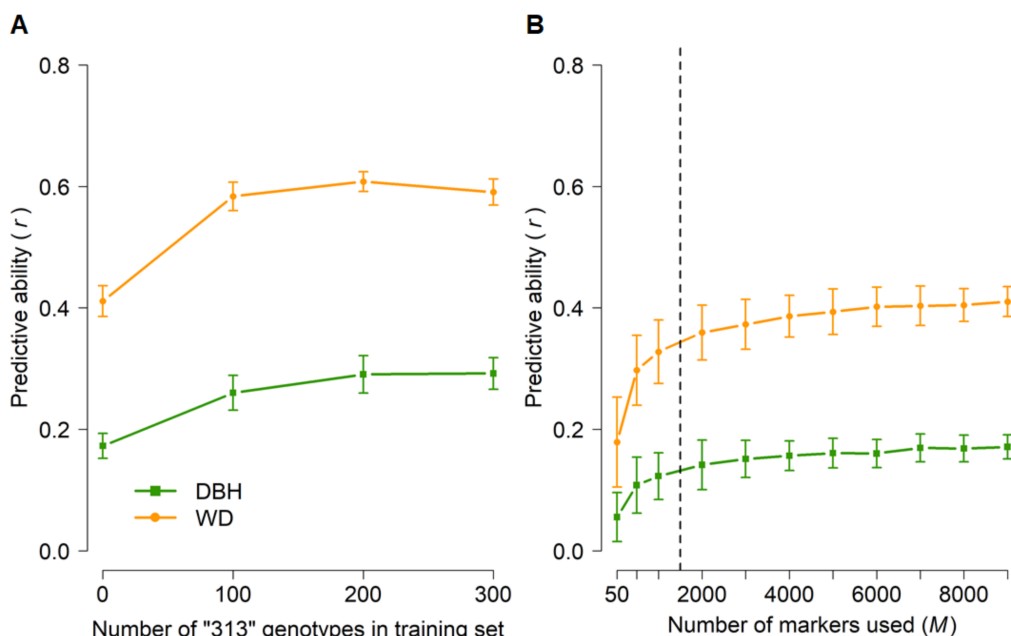

**Figure 5.** Genomic predictive abilities across populations for diameter at breast height (DBH) and wood density (WD) as a function of the number of "313" genotypes included in the training set (**A**) and the number of markers (**B**) used in cross-validations across populations. Error bars correspond to standard deviations across 100 iterations for each set of parameters. For analyses in (**A**), a training set of 800 genotypes (including the desired number of "313" genotypes) was randomly sampled from the total number of 1761 genotypes (Table 1) and all 9353 SNPs were used. The training set in (**B**) was selected at random ($N = 800$), with no "313" genotypes included, and subsets of markers were selected at random. All predictive abilities were only for "313" genotypes that were not included in the training set ($N = 381–681$, see Materials and Methods and Table 1). The dashed line in (**B**) indicates the plateau in predictive ability.

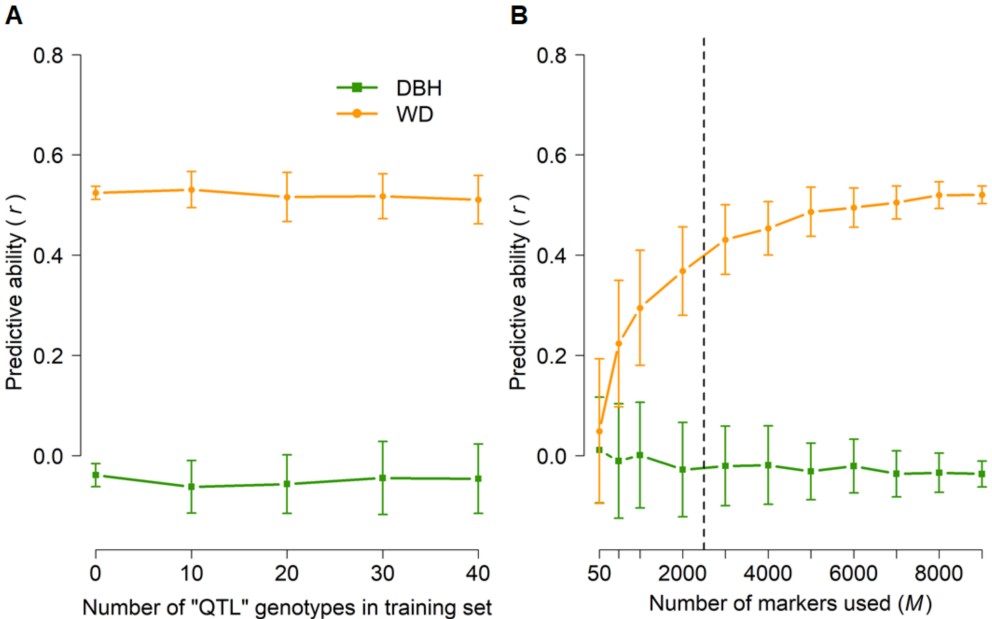

**Figure 6.** *Cont.*

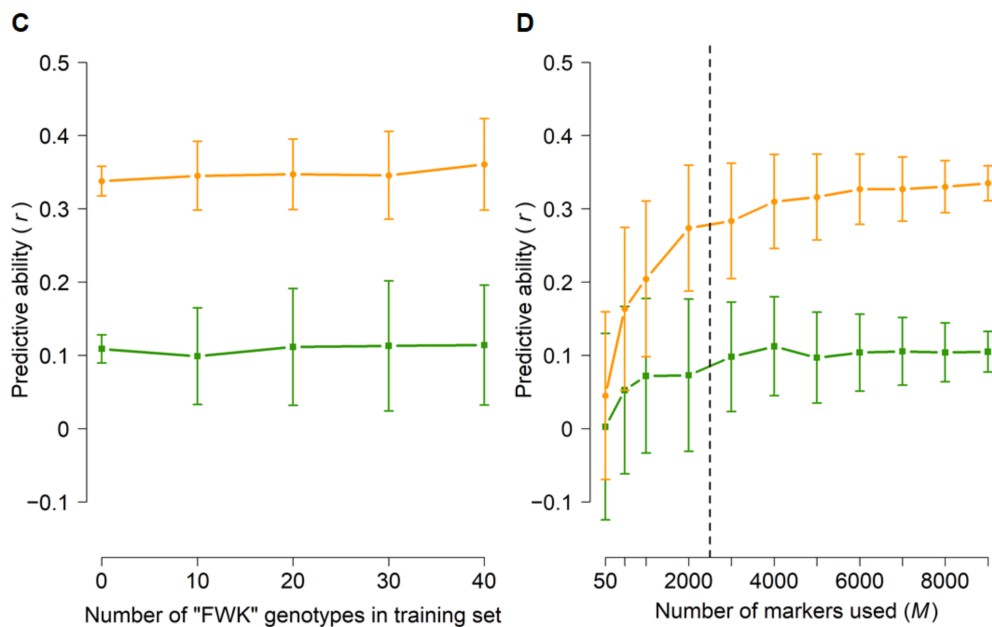

**Figure 6.** Genomic predictive ability within families. GBLUP predictive ability for diameter at breast height (DBH) and wood density (WD) as a function of the number of "QTL" or "FWK" genotypes included in the training set (**A**,**C**) and the number of markers (**B**,**D**) used in cross-validations. Error bars correspond to standard deviations across 100 iterations for each set of parameters. For analyses in (**A**,**C**), a training set of 1600 genotypes (including the desired number of "QTL" or "FWK" genotypes) was randomly sampled, and all 9353 markers were used. The training set in (**B**,**D**) was selected at random (*N* = 1600), with no "QTL" or "FWK" genotypes included, and subsets of markers were selected at random. All predictive abilities were only for "QTL" or "FWK" genotypes that were not included in the training set (*N* = 41–86, see Materials and Methods and Table 1). The dashed lines in (**B**,**D**) indicate the plateaus in predictive ability.

### 3.4. Pedigree-Based and Single-Step Blended Prediction (ABLUP and HBLUP)

The main objective of these analyses was to assess the potential of performing predictions across series of trials, only some of which are genotyped (Table 1). The results obtained from single-step blended prediction (HBLUP) were generally consistent with those from genomic prediction (GBLUP, Figures 4–6) for both phenotypic traits, but the differences in predictive ability between models and populations were more pronounced for DBH (Tables 3 and 4). The predictive ability for DBH in the "Cloned Elites" series was nearly doubled (87% increase) using the best HBLUP model (*r* = 0.227) compared to the traditional ABLUP analysis with the originally recorded pedigree (*r* = 0.121). This clearly reflects the impact of genomic data in both improving the accuracy of pedigree information and capturing "hidden" relationships not reflected in pedigree records. While this pattern was generally consistent across trial series, several factors affected the performance of ABLUP and HBLUP, particularly for DBH (Tables 3 and 4).

**Table 3.** Predictive abilities for pedigree-based (ABLUP) versus single-step blended (HBLUP) prediction analyses for diameter at breast height (DBH). Results from the best HBLUP model are shown in bold. Cross-validations were performed by including all but one trial series in the training set, with the remaining trial series used as a test set (see Materials and Methods).

| Pedigree Option | Trial Series | ABLUP | HBLUP—Weight on Pedigree Information | | | | | |
|---|---|---|---|---|---|---|---|---|
| | | | **0.05** | **0.1** | **0.2** | **0.3** | **0.4** | **0.5** |
| Originally recorded | FR260 | 0.227 | 0.221 | 0.221 | 0.221 | 0.223 | 0.224 | **0.224** |
| | FR305GF | 0.441 | 0.382 | 0.391 | 0.404 | 0.414 | 0.421 | **0.428** |
| | FR305HD | 0.459 | 0.497 | 0.498 | **0.499** | 0.499 | 0.498 | 0.497 |
| | FR353 | 0.241 | **0.334** | 0.329 | 0.320 | 0.312 | 0.304 | 0.297 |
| | Cloned Elites | 0.121 | **0.200** | 0.199 | 0.199 | 0.198 | 0.196 | 0.193 |
| Corrected, unresolved relationships as originally recorded | FR260 | 0.221 | 0.215 | 0.216 | 0.217 | 0.218 | 0.218 | **0.219** |
| | FR305GF | 0.402 | 0.373 | 0.381 | 0.391 | 0.399 | 0.404 | **0.409** |
| | FR305HD | 0.460 | 0.497 | 0.497 | **0.498** | 0.498 | 0.497 | 0.496 |
| | FR353 | 0.241 | **0.333** | 0.330 | 0.324 | 0.319 | 0.313 | 0.306 |
| | Cloned Elites | 0.151 | 0.208 | 0.210 | 0.212 | **0.213** | 0.212 | 0.210 |
| Corrected, unresolved relationships set as "unknown" | FR260 | 0.227 | 0.214 | 0.216 | 0.217 | 0.218 | 0.220 | **0.221** |
| | FR305GF | 0.382 | 0.367 | 0.372 | 0.379 | 0.384 | 0.389 | **0.393** |
| | FR305HD | 0.462 | 0.496 | 0.498 | 0.499 | **0.500** | 0.500 | 0.498 |
| | FR353 | 0.274 | **0.332** | 0.331 | 0.328 | 0.326 | 0.323 | 0.320 |
| | Cloned Elites | 0.195 | 0.209 | 0.213 | 0.219 | 0.223 | 0.226 | **0.227** |

**Table 4.** Predictive abilities for pedigree-based (ABLUP) versus single-step blended (HBLUP) prediction analyses for wood density (WD). Results from the best HBLUP model are shown in bold. Cross-validations were performed by including all but one trial series in the training set, with the remaining trial series used as a test set (see Materials and Methods).

| Pedigree Option | Trial Series | ABLUP | HBLUP—Weight on Pedigree Information | | | | | |
|---|---|---|---|---|---|---|---|---|
| | | | **0.05** | **0.1** | **0.2** | **0.3** | **0.4** | **0.5** |
| Originally recorded | FR260 | 0.232 | **0.294** | 0.291 | 0.287 | 0.283 | 0.280 | 0.276 |
| | FR305GF | 0.413 | 0.390 | 0.400 | 0.412 | 0.419 | 0.426 | **0.431** |
| | FR305HD | 0.396 | 0.411 | 0.415 | 0.419 | 0.421 | 0.421 | **0.421** |
| | FR353 | 0.318 | 0.431 | 0.435 | **0.438** | 0.437 | 0.435 | 0.430 |
| | Cloned Elites | 0.371 | 0.402 | 0.409 | 0.420 | 0.430 | 0.438 | **0.444** |
| Corrected, unresolved relationships as originally recorded | FR260 | 0.316 | 0.304 | 0.305 | 0.306 | 0.306 | 0.306 | **0.307** |
| | FR305GF | 0.412 | 0.389 | 0.400 | 0.413 | 0.420 | 0.425 | **0.428** |
| | FR305HD | 0.365 | 0.413 | **0.414** | 0.413 | 0.412 | 0.409 | 0.406 |
| | FR353 | 0.331 | 0.441 | 0.445 | **0.449** | 0.449 | 0.447 | 0.442 |
| | Cloned Elites | 0.392 | 0.403 | 0.409 | 0.418 | 0.426 | 0.432 | **0.438** |
| Corrected, unresolved relationships set as "unknown" | FR260 | 0.334 | 0.327 | 0.328 | 0.329 | 0.329 | 0.330 | **0.330** |
| | FR305GF | 0.415 | 0.369 | 0.380 | 0.393 | 0.402 | 0.408 | **0.412** |
| | FR305HD | 0.367 | **0.411** | **0.411** | 0.410 | 0.409 | 0.407 | 0.405 |
| | FR353 | 0.362 | 0.438 | 0.442 | 0.446 | **0.447** | 0.445 | 0.442 |
| | Cloned Elites | 0.427 | 0.400 | 0.406 | 0.416 | 0.425 | 0.433 | **0.440** |

First, pedigree accuracy was critical. Using corrected pedigree information tended to result in higher ABLUP predictive ability for genotyped trial series, and particularly for the "Cloned Elites" series, where a large number of pedigree inconsistencies were detected. Setting unresolved relationships as "unknown" tended to be the best approach (ABLUP columns in Tables 3 and 4), although this was not always true. Second, SNP data tended to increase predictive abilities beyond the effect of using corrected pedigree information (i.e., ABLUP versus HBLUP columns in Tables 3 and 4). Third, the degree of relatedness between the training and test populations largely determined predictive ability. For example, the lowest predictive ability was for the "Cloned Elites" because they

have relatively weak relatedness to the populations in the "FR353" and "FR305" series. In contrast, predictive ability was much higher for the "FR305" series, presumably because of its higher relatedness to populations in the "FR260" and "FR353" series. Fourth, the quality of phenotypic data might have played a role. For example, the predictive ability for the "FR260" series was relatively low, possibly because phenotypes from that series were based on a single observation (i.e., no clonal replication). This effect, however, may be confounded with the lack of genotypic data for the "FR260" series. Finally, the optimal weighting of pedigree versus SNP information differed considerably among trial series and HBLUP scenarios. As expected, where (i) genotyping data were absent, (ii) there were suspected mismatches between phenotypic and SNP records, or (iii) high rates of pedigree errors had been resolved (i.e., in the "FR260", "FR305", and "Cloned Elites" series, respectively), the optimal weighting of pedigree data was high (0.5). In contrast, the optimal weighting for the "Cloned Elites" was very low (0.05) prior to resolving pedigree errors (Tables 3 and 4).

## 4. Discussion

Recent advances with DNA sequencing and genotyping technology [53] have enabled studies of both long-term evolutionary history [54,55] and recent demographic events [56], as well as the recovery of missing genealogy in pedigreed populations and the prediction of unmeasured phenotypes [57,58]. Furthermore, while results from early genome-wide association studies were likely limited by low statistical power and confounding caused by population structure [23,59–61], this field is gathering momentum as data sets are becoming larger [16–19]. Although the dissection of complex trait genetic architecture remains an aspirational goal for the medium to long term, our results clearly show that genomic data can be practically useful in tree improvement in the short term by enabling the (1) improvement of pedigree records and control of inbreeding; (2) prediction of phenotypes from genomic data (GBLUP); and (3) integration of historical and contemporary phenotypic and genomic data sets through single-step blended prediction (HBLUP). In the following sections, we discuss our results in the context of each of these potential applications, highlighting outstanding issues and caveats.

### 4.1. Population Structure, Pedigree Reconstruction, and Control of Inbreeding

Tree improvement programmes globally are not very advanced because of the long generation cycles of even the fastest-growing species [62,63]. However, management of inbreeding may be necessary even after only three breeding cycles [64] and will inevitably become important in the future. Genomic information provides a wide range of options for maintaining diversity and avoiding inbreeding depression [14,65]. Correct pedigree information is critical for the accurate estimation of breeding values, other genetic parameters [66,67], and genetic gain [68]. Pedigree reconstruction is a powerful tool for correcting pedigree record errors [64,69]. Unsampled parental candidates can make pedigree reconstruction difficult, but sib-ship reconstruction can also be used [70]. Furthermore, relatedness among candidate parents introduces additional challenges for reconstructing pedigrees, even when thousands of genetic markers are used. Our results confirmed these challenges. Statistically significant parentage could not be assigned for most offspring for which pedigree inconsistencies were uncovered, presumably because the true parents had not been genotyped. We compared two approaches to dealing with such cases: either keeping the originally recorded parent or setting pedigree records to "parent unknown". Our ABLUP and HBLUP results (Tables 3 and 4) supported the intuitive expectation that allowing missing data is better than using incorrect data. Finally, while ABLUP and HBLUP predictive abilities were generally improved by using corrected pedigrees (Tables 3 and 4), changes in heritability were more variable (Table 2). Thus, the presumed increase in genetic gain as a result of using more accurate pedigree records is yet to be demonstrated in radiata pine.

### 4.2. Genomic Prediction (GBLUP)

An important measure of the success of breeding programmes is the genetic gain achieved per unit of time. Thus, opportunities for shortening breeding cycles are appealing [8]. However, this can be particularly challenging in forest species where many economically important traits are expressed at a later age, preventing early selection. The ability to predict unobserved phenotypes through a genome-wide set of molecular markers (i.e., genomic selection) is a promising strategy for overcoming this challenge [23,41].

Through various re-sequencing and array-based approaches, medium- to high-density SNP genotyping platforms have recently become available for several forest tree species [16,31,71–73]. Theoretically, this allows capturing genetic relatedness, Mendelian segregation [74], and, possibly, linkage disequilibrium between markers and causative loci in genomic prediction [75], compared to only expected levels of relatedness in pedigree-based analyses. A seminal simulation study by Grattapaglia and Resende [76] highlighted the potential of genomic selection in forest tree species. Empirical studies quickly ensued, using both open-pollinated [20,21] and control-pollinated families [71], as well as genotyping platforms ranging from genome-wide GBS [20,21] to exome capture [22] and custom SNP arrays [77,78]. As predicted by theoretical considerations, simulations, and well-characterised case studies [79], the first generation of empirical studies in forest trees showed that the level of relatedness between the training and test populations is by far the most important determinant of predictive ability. Our results were consistent with this trend. Random cross-validations (Figure 4), predictions across populations separated by one generation (Figure 5), and predictions within untested families (Figure 6) consistently showed that genomic prediction should be achievable for moderately heritable traits such as WD. Furthermore, genomic prediction may also work for more challenging traits such as DBH, provided that the training population is large enough (ideally $\geq 1000$) and contains many close relatives (e.g., the parents) of the individuals in the prediction population.

It is not clear whether the predictive abilities we observed would persist across multiple generations. Linkage disequilibrium between markers and causative loci is believed to be the only persistent component of genomic prediction across generations [80]. Capturing this linkage disequilibrium at the population level will be particularly challenging in organisms with complex genomes, containing a large proportion of repetitive elements (such as radiata pine and most other commercially important conifers) or higher levels of ploidy [81]. The implementation of multi-generation prediction models might help with mitigating this issue [82], but the marker density (and associated cost) required to assure robust genomic prediction across generations is currently unknown. In our analyses, as few as 2000–5000 markers were sufficient to reach a plateau in predictive ability, which is not surprising given the high level of relatedness within and across the populations we used and is consistent with both theoretical [75] and empirical results [71]. It is also possible that linkage disequilibrium caused by the recent admixture of breeding populations developed in New Zealand, including the surprisingly high proportion of trees with island ancestry [32], made it possible to achieve moderate GBLP predictive abilities with a relatively small number of markers. Genomic prediction in more homogeneous populations of distantly related individuals indicate that the number of markers needed to saturate predictive ability is both trait- and population-specific [47,83,84].

Finally, within-family prediction is likely to be the crux of the operational implementation of genomic selection. Based on our results, within-family predictive abilities were moderate for WD but probably too low to be considered operationally viable for DBH. Additional empirical experiments, with larger numbers of inter-connected families, are currently in progress. The results from these experiments, as well as the cost of DNA extraction and genotyping, will likely drive the rate of operational implementation of genomic selection in the short to medium term (i.e., 2–5 years).

*4.3. Single-Step Blended Prediction (HBLUP)*

We demonstrated the potential impact of genomic data in two ways. First, both traditional ABLUP and blended HBLUP predictive abilities tended to improve when a genomics-corrected pedigree was used and unresolved relationships were set as "unknown". This is probably because we used an advanced generation breeding population with a complex pedigree, and any incorrect parentage assignments would have had cascading effects of incorrect genetic relationships and potentially affected the matrix rescaling step in HBLUP [85]. Second, in addition to the effect of using corrected pedigree information, SNP data tended to further improve predictive abilities in HBLUP analyses, compared to traditional ABLUP analyses (Tables 3 and 4). This was presumably because genomic data improved the estimation of the true relationships among trees and uncovered relationships that were not reflected in the pedigree records. Furthermore, differential weighting of pedigree versus genomic information in HBLUP analyses provided further insights about (1) the importance of the relationship between the training and test populations; (2) potential labelling issues leading to mismatches between phenotypic and SNP records; and (3) the relative frequency of resolved pedigree errors in different populations. Therefore, careful consideration should be given to the composition of the training population and trait heritability [86], so that the optimal weighting of pedigree information remains consistent and robust [87,88].

## 5. Conclusions

Our study clearly illustrates the potential benefits of using genome-wide molecular marker information in tree improvement. First, using SNP markers, we substantially improved the accuracy of pedigree information, leading to increased predictive abilities. Second, genomic prediction was possible across populations for both phenotypic traits included in this study (DBH and WD). Third, within-family genomic prediction was possible for the more highly heritable WD trait using moderately large training populations ($N$ = 800–1200) and as few as 2000–5000 SNP markers. Finally, we demonstrated that single-step blended prediction combining historical and contemporary data sets can be effective. However, the practical significance of our results still needs to be quantified in empirical case studies as it is not yet clear how the promising predictive abilities we observed will translate into genetic gain per unit of time and whether genotyping costs are justified. Therefore, the rate of operational implementation of genomic or single-step blended prediction over the next few years will be driven by direct empirical evidence of economic feasibility.

**Supplementary Materials:** The following supporting information can be downloaded at: https://www.mdpi.com/article/10.3390/f13020282/s1, Figure S1: Distribution of identity-by-state (IBS) coefficients using different genotyping platforms. Table S1: Description of field experiments used in single-step genomic evaluation. Table S2: Simulation-based evaluation of parentage assignment using the *apparent* R package.

**Author Contributions:** Conceptualisation: J.K., G.T.S. (Gancho T. Slavov), M.P. and H.S.D.; Methodology/Formal Analysis: J.K., A.I. and G.T.S. (Gancho T. Slavov); Resources: N.J.G., G.T.S. (Grahame T. Stovold) and M.P.; Writing: J.K., A.I., N.J.G. and G.T.S. (Gancho T. Slavov). All authors have read and agreed to the published version of the manuscript.

**Funding:** The study was funded by the New Zealand Ministry of Business, Innovation and Employment (MBIE) 2019 Partnered Research Fund in Forest Genetics (contract number C04X1808) and through a Collaboration Agreement between the RPBC and Scion.

**Data Availability Statement:** The data presented in this study are available on request from the corresponding author. The data are not publicly available due to their commercial sensitivity for the RPBC.

**Acknowledgments:** We thank Glenn Howe (Oregon State University), Alec Foster (RPBC), and Rowland Burdon, Michael Watt, and David Pont (Scion) for their comments and suggestions on earlier versions of this manuscript.

**Conflicts of Interest:** The authors declare no conflict of interest. The funders had no role in the design of the study; in the collection, analyses, or interpretation of data; in the writing of the manuscript; or in the decision to publish the results.

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
