# Peer review of "Genomics-Enabled Management of Genetic Resources in Radiata Pine"

_forests, doi:10.3390/f13020282_

Round 1

Reviewer 1 Report

This research performed the pedigree reconstruction analysis using two phenotypic datasets and abundant genomic SNP data for five radiata pine populations (or four? because none sample from population 260 was genotyped). Predicted models were compared based on the two data types, and the results suggested potential advantages of genomic resources in the enhancement of tree improvement in pine species. The methodology and results in the research are efficient and provide positive evidences. Some questions were listed below for the research: line 86: did the subset trees from the five populations come from the five natural locations of the radiata pine in USA and Mexico, respectively? line 91: is there any reason to measure the DBH at the tree height of 1.4 metre? line 283: One possible explanation for the inconsistent definition of individuals may be the complex intraspecific gene flow/introgression among populations of the radiata pine. This aspect is evidenced by the PCA result in figure 2, which showed high mixture of individuals from the four genotyped populations. line 383: delete one "the" Table 1: for the description of the two parent populations QTL and FWK, what is the meaning of the numbers in parentheses? Table 2: the "ABLUP" model could not be found in the METHOD part, is it GBLUP? As the cross-validations were used for model simulation, could the prediction provide statistical significance for the estimated narrow-sense heritability index? Figures 5,6: DBH showed lower predictive ability than WD in all figures, probably due to the local adaptaion of trees in the face of different environments that might not related to genetic regulation from parents, while WD might be a heritable trait in the breeding system.

Author Response

We thank Reviewer 1 for their helpful comments, suggestions, and questions. Below are our specific answers to each question.

“line 86: did the subset trees from the five populations come from the five natural locations of the radiata pine in USA and Mexico, respectively? “

All populations included in this study were derived from the radiata pine breeding programme in New Zealand. These were believed to have been derived primarily from the mainland provenances in California. However, recent evidence suggests that admixture with island provenances from Mexico may be more extensive than previously recognised (see companion paper by Graham et al., reference [32]). We therefore acknowledged admixture-caused linkage disequilibrium as a possible contributing factor to genomic predictive abilities (section 4.2).

line 91: is there any reason to measure the DBH at the tree height of 1.4 metre?

This is standard practice in New Zealand forestry (e.g., https://mpi.govt.nz/dmsdocument/45-indigenous-forestry-measuring-indigenous-trees-and-logs-a-field-guide, https://www.nzffa.org.nz/system/assets/5854/p20044coll6_21.pdf). We have clarified this in section 2.1.

“line 283: One possible explanation for the inconsistent definition of individuals may be the complex intraspecific gene flow/introgression among populations of the radiata pine. This aspect is evidenced by the PCA result in figure 2, which showed high mixture of individuals from the four genotyped populations.”

We have acknowledged admixture-caused linkage disequilibrium as a possible contributing factor to genomic predictive abilities in section 4.2.

“line 383: delete one "the" “

We have made this correction.

“Table 1: for the description of the two parent populations QTL and FWK, what is the meaning of the numbers in parentheses?”

The numbers in parenthesis represent the number of crosses made using the respective parents. This also applies to the “QTL” and “FWK” biparental mapping families.

“Table 2: the "ABLUP" model could not be found in the METHOD part, is it GBLUP? As the cross-validations were used for model simulation, could the prediction provide statistical significance for the estimated narrow-sense heritability index?”

We have included more detailed description of the ABLUP methodology, including for the estimation of standard errors of narrow-sense heritabilities in Table 2. All narrow-sense heritabilities in Table 2 were statistically significant.

“Figures 5,6: DBH showed lower predictive ability than WD in all figures, probably due to the local adaptation of trees in the face of different environments that might not related to genetic regulation from parents, while WD might be a heritable trait in the breeding system.”

The question about the relative roles of DBH and WD in local adaptation is interesting, but outside the scope of this manuscript. The differences in heritability are discussed in section 3.3. We tried to make this point clearer in the revised version of the manuscript.

Reviewer 2 Report

Klapste et al provide a clearly written manuscript exploring the use of genomic selection in a radiata pine breeding program. They showed that genomic information (even partial information) in an ssBLUP framework provides additional predictive ability for traits (in this case, diameter or wood density) in many families, depending on population size and relatedness.

I have only few minor suggestions.

To interpret Tables 3 and 4 and I had to go back to the methods, which specify “Cross-validations were performed by including all but one trial series in the training population, with the remaining trial series used as a test population.”  I would restate that in the results and/or table legend. to clear up any confusion

Rare typos:

Figure 3: “withing”

Page 12: “the the” repeat in “For example, the the lowest predictive ability was for the “Cloned Elites” as they have relatively poor relatedness to the populations in the “FR353” and “FR305” series.”

Author Response

We thank Reviewer 1 for their helpful suggestions. Below are our specific responses to these suggestions.

“To interpret Tables 3 and 4 and I had to go back to the methods, which specify “Cross-validations were performed by including all but one trial series in the training population, with the remaining trial series used as a test population.”  I would restate that in the results and/or table legend. to clear up any confusion”

We added this sentence to the legends of Tables 3 and 4.

“Rare typos: Figure 3: “withing” Page 12: “the the” repeat in “For example, the the lowest predictive ability was for the “Cloned Elites” as they have relatively poor relatedness to the populations in the “FR353” and “FR305” series.”

We have corrected both of these typos.